# Phenotypic plasticity evolves at multiple biological levels in response to environmental predictability in a long-term experiment with a halotolerant microalga

**Christelle Leung** [1]*, **Daphné Grulois**[1], **Leandro Quadrana**[2], **Luis-Miguel Chevin**[1]*

**1** CEFE, Université de Montpellier, CNRS, EPHE, IRD, Montpellier, France, **2** Institut of Plant Science Paris-Saclay (IPS2), UMR Université Paris Saclay—CNRS 9213—INRAE 1403, Université Evry Val d'Essonne—Université Paris Diderot, Paris, France

\* christelle.leung@umontreal.ca (CL); luis-miguel.chevin@cefe.cnrs.fr (L-MC)

**Data Availability Statement:** Raw sequence data (RNA-seq and WGB-seq) used in this study are deposited in the NCBI's Sequence Read Archive

## Abstract

Phenotypic plasticity, the change in the phenotype of a given genotype in response to its environment of development, is a ubiquitous feature of life, enabling organisms to cope with variation in their environment. Theoretical studies predict that, under stationary environmental variation, the level of plasticity should evolve to match the predictability of selection at the timing of development. However, the extent to which patterns of evolution of plasticity for more integrated traits are mirrored by their underlying molecular mechanisms remains unclear, especially in response to well-characterized selective pressures exerted by environmental predictability. Here, we used experimental evolution with the microalgae *Dunaliella salina* under controlled environmental fluctuations, to test whether the evolution of phenotypic plasticity in responses to environmental predictability (as measured by the squared autocorrelation $\rho^2$) occurred across biological levels, going from DNA methylation to gene expression to cell morphology. Transcriptomic analysis indicates clear effects of salinity and $\rho^2$ × salinity interaction on gene expression, thus identifying sets of genes involved in plasticity and its evolution. These transcriptomic effects were independent of DNA methylation changes in *cis*. However, we did find $\rho^2$-specific responses of DNA methylation to salinity change, albeit weaker than for gene expression. Overall, we found consistent evolution of reduced plasticity in less predictable environments for DNA methylation, gene expression, and cell morphology. Our results provide the first clear empirical signature of plasticity evolution at multiple levels in response to environmental predictability, and highlight the importance of experimental evolution to address predictions from evolutionary theory, as well as investigate the molecular basis of plasticity evolution.

## 1. Introduction

Phenotypic plasticity, the ability of a given genotype to produce different phenotypes depending on environmental conditions, is an important mechanism enabling organisms to cope

(SRA) database under BioProject PRJNA736997 (https://www.ncbi.nlm.nih.gov/bioproject/?term= PRJNA736997). The specific samples used in this study are under the BioSample accessions listed in S1 and S2 Tables. DNA methylation 100 bp tiles percentage values, raw RNA-seq read counts (i.e. transcript expression values), structural and functional gene annotations, and cell morphology data have been deposited in Figshare https://doi. org/10.6084/m9.figshare.21905670.

**Funding:** This work was supported by the European Research Council (Grant 678140-FluctEvol) to LMC, a Fonds de Recherche du Québec - Nature et Technologies (FRQNT) postdoctoral fellowship to CL, and a travel grant for collaboration provided by the GDR Plasticité Phénotypique (GDR 3715) from CNRS (CL). The funders had no role in study design, data collection and analysis, decision to publish, or preparation of the manuscript.

**Competing interests:** The authors have declared that no competing interests exist.

**Abbreviations:** BH, Benjamini–Hochberg; DE, differentially expressed; DMR, differentially methylated region; FSC, Forward Scatter; GO, Gene Ontology; LRT, likelihood ratio test; RDA, redundancy analyses; RNA-seq, RNA-sequencing; SSC, Side Scatter; TSS, transcription start site; WGBS, whole-genome bisulfite sequencing.

with variation in their environment. Understanding what drives the evolution of plasticity, from selective causes to underlying mechanisms, is thus important not only for basic research, but also for on our ability to predict the fate of populations, especially under global change [1,2]. Despite this interest, we still lack critical empirical information on the extent to which selection on phenotypic plasticity propagates across hierarchical levels of the organisms, from more integrated phenotypes that are directly exposed to selection, to their underlying molecular basis.

Theoretical and empirical studies have demonstrated that the adaptiveness of phenotypic plasticity arises from its interplay with environmental variation in selection, when the latter is partly predictable [3–7]. A high degree of plasticity is expected to be favored when the environmental cue causing induction of a given phenotype is a reliable predictor of selection acting on this phenotype. Conversely, under unpredictable environmental changes (or unreliable cues), plasticity leads to phenotypes that are often mismatched with their optimum, such that lower degrees of plasticity are selected [3–7]. In the limit of completely unpredictable environments, bet-hedging strategies that do not rely on environmental cues may even be selected [8,9].

In terms of mechanisms and pathways, the increased availability of -omics methods has greatly facilitated our understanding of the relationship between molecular changes and changes in phenotypic expression. In particular, many studies have shown that phenotypic plasticity often involves at its most basic level variation in gene expression with the environment [10–14]. This suggests that the molecular basis of phenotypic plasticity should also encompass mechanisms that regulate gene expression, as clearly demonstrated in a number of cases [12–14]. In particular epigenetic processes, i.e., modifications of chromatin beyond DNA sequence, transmitted through mitosis (and possibly meiosis), and potentially influencing gene expression, were shown to be important molecular mechanisms for phenotypic plasticity [15–18]. As both gene expression and epigenetic marks (such as DNA methylations) are known to be under genetic control [19–21], selection on phenotypic plasticity for more integrated traits is thus expected to cascade down to cause evolution of plasticity for its underlying molecular mechanisms (e.g., [22]), but this process has seldom been investigated experimentally. More specifically, the extent to which classical theoretical predictions about evolution of plasticity in response to environmental predictability hold from molecular phenotypes to more integrated ones remains unexplored.

Recently, we have shown that environmental variation in CpG methylation and gene expression in the halotolerant microalga *Dunaliella salina* contributes to phenotypic plasticity in this species [23]. In addition, we confirmed the contribution of the genotype to salinity responses at the levels of DNA methylation, gene expression, and population growth rate, highlighting the evolutionary potential of phenotypic plasticity at multiple levels [23]. Importantly, long-term experimental evolution in this species has demonstrated that reduced plasticity in cell shape and content evolved in populations confronted to less predictable environments [5]. Here, we used experimental evolution under controlled environmental fluctuations to assess whether the evolution of plasticity in responses to environmental predictability permeates across biological levels, from DNA methylation to gene expression to more integrated phenotypes.

## 2. Results

We analyzed 9 populations of the halotolerant microalga *D. salina* (strain CCAP 19/15) that have evolved under regimes of randomly fluctuating environments, with controlled and variable predictability [5,24]. During experimental evolution, lines derived from a single ancestral population were exposed to randomly fluctuating salinity, with changes every 3 or 4 days (with

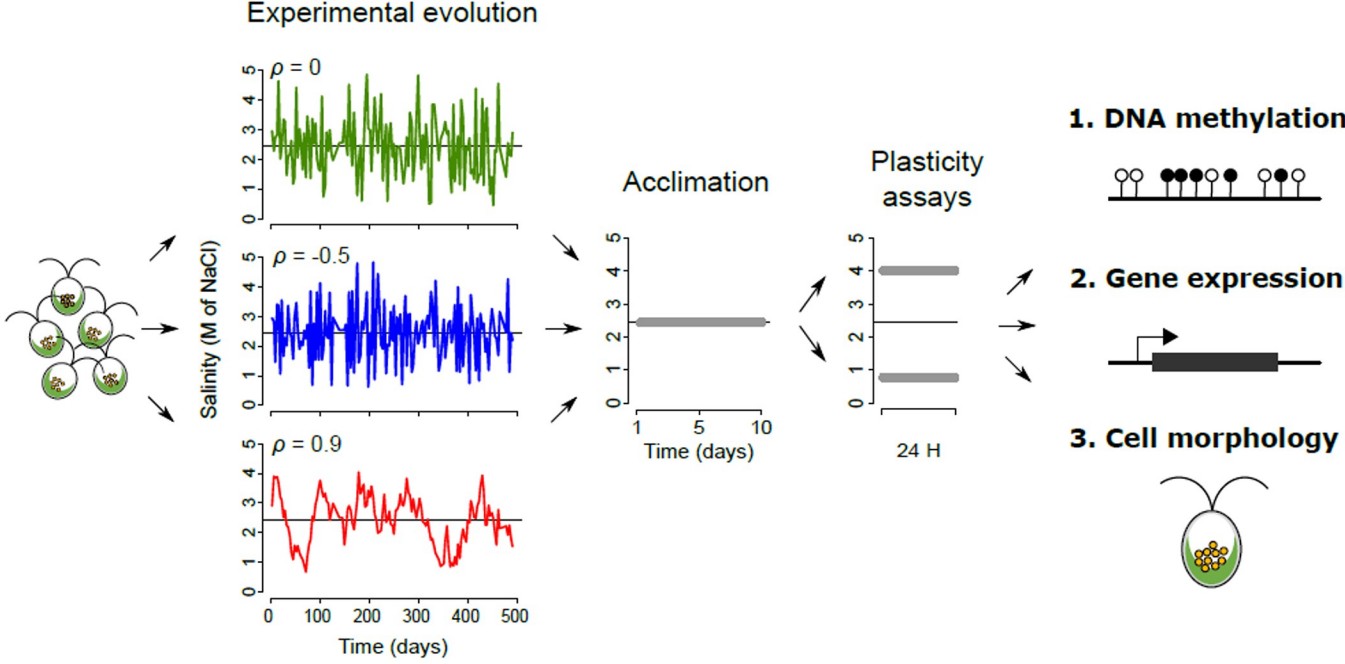

**Fig 1. Experimental design.** The experiment included 3 steps: (i) long-term experimental evolution (left); (ii) acclimation at a constant, intermediate salinity (middle); and (iii) plasticity assays at high versus low salinity (right). Each colored time series on the left represents an actual realization of salinity fluctuations for one of the populations used in this study, with the color denoting the treatment of stationary (i.e., expected long-term) temporal autocorrelation ($\rho$). At the end of the experiment, cells were harvested for DNA methylation, gene expression, and cell morphology analyses.

about one generation per day), for a total of ca. 500 generations. Salinity had a normal distribution over time, with the same mean (2.4 M NaCl) and standard deviation (1 M NaCl) across treatments, but variable autocorrelation, and, hence, variable predictability [5,24]. Previous morphological analysis of 32 of these lines revealed that reduced morphological plasticity has evolved in lines that experienced less predictable environments [5]. To further characterize the molecular basis of this evolution, we set out to determine whether the plasticity of DNA methylation and gene expression levels evolved for a subset of these lines that experienced 3 different predictability treatments (3 lines per treatment): low ($\rho^2 = 0$), intermediate ($\rho^2 = 0.25$), and high ($\rho^2 = 0.81$) predictability, where $\rho$ is the stationary (long-term) temporal autocorrelation of salinity time series. All populations were subsequently subjected to a 10-day acclimation step at intermediate salinity ([NaCl] = 2.4 M), to ensure they had similar physiological states and population densities before the phenotypic and molecular plasticity assays. They were then placed for 24 h at 2 salinities near the extremes of their historical range ([NaCl] = 0.8 M and 4.0 M), to assess their degree of plasticity in DNA methylation, gene expression, and individual cell morphology (Fig 1). At any of these levels, an effect of salinity is indicative of phenotypic plasticity, an effect of the evolutionary treatment $\rho^2$ denotes evolution, while $\rho^2 \times$ salinity interaction indicates evolution of plasticity.

## 2.1 DNA methylation and gene expression plasticity evolved in response to environmental predictability

DNA methylation can contribute to phenotypic differentiation by influencing gene expression regulation [18, 25–28]. To assess whether experimental evolution of *D. salina* may lead to epigenetic differentiation, we performed whole-genome bisulfite sequencing (WGBS) for all samples, yielding a total of $1.23 \times 10^9$ 150 bp paired-end raw reads, and an estimated average depth

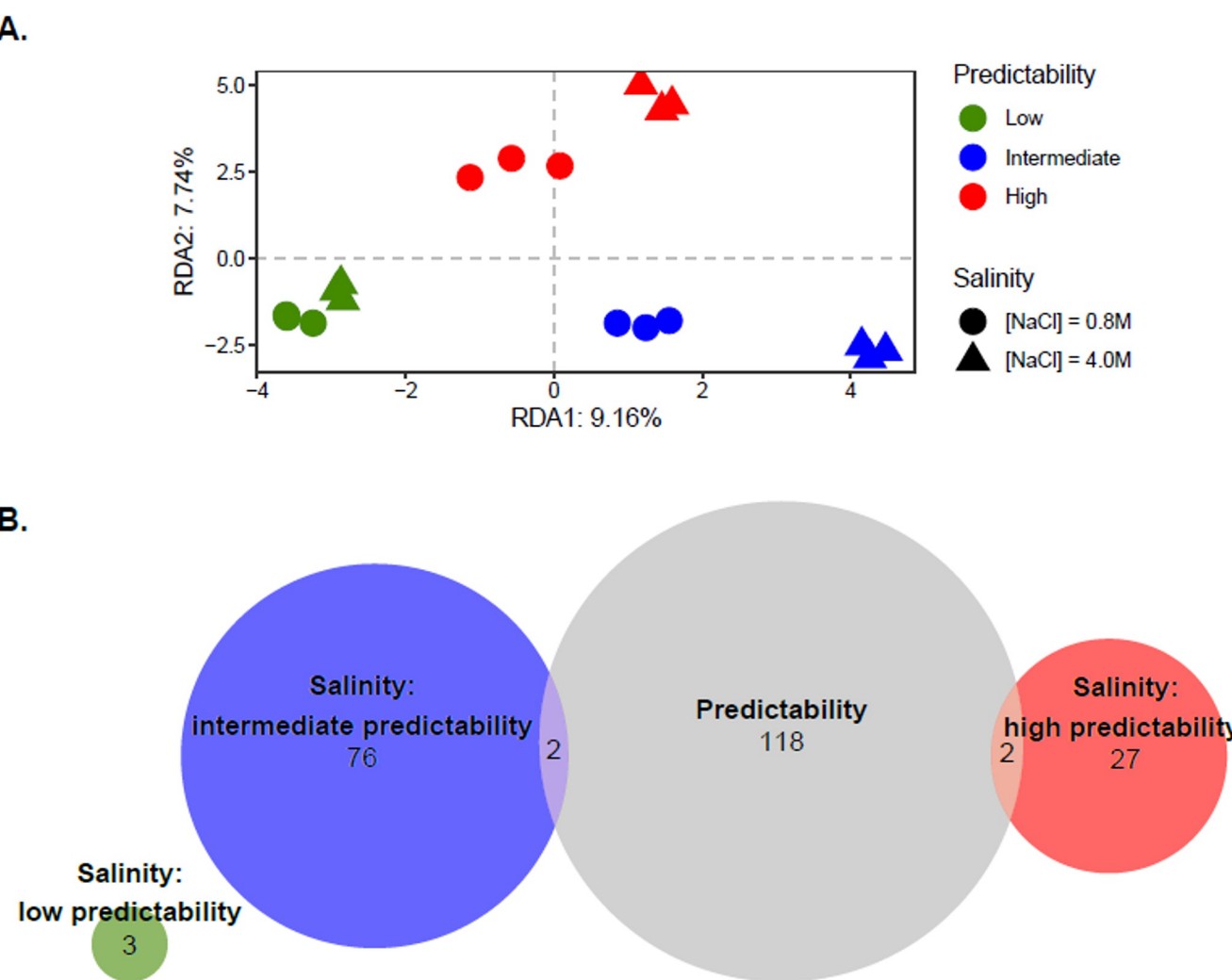

**Fig 2. Evolution and plasticity of DNA methylation. (A)** Variation of DNA methylation patterns across evolved populations. RDA plot performed on DNA methylation patterns according to evolution conditions (colors) and salinity during plasticity assay (shapes). **(B)** Distribution of DMRs. Venn diagram describing the number of DMRs ($q$-value < 0.05 and |diff-Methylation| > 20%) across evolutionary conditions (Predictability, light gray), and between salinities within each of the 3 evolutionary conditions: low (green), intermediate (blue), and high (red) predictability of environmental changes. The raw data underlying this figure are available in the Figshare repository https://doi.org/10.6084/m9.figshare.21905670. DMR, differentially methylated region; RDA, redundancy analysis.

of coverage of 43.76 × (*s.d.* 3.14 ×) per sample (S1 Table). After data filtering, we carried out our methylation analyses on an average of $7.56 \times 10^7$ (*s.d.* $8.28 \times 10^6$) cytosines per samples at the CpG context (S1 Table), where methylations are most prevalent in this species [23]. Redundancy analyses (RDA) based on overall CpG methylation revealed a significant effect of evolutionary treatments $\rho^2$ ($R^2_{adj}$ = 4.66%; $P$ = 0.005) on DNA methylation (Fig 2A). Fig 2A, which is a constrained ordination that maximizes variation in DNA methylation that can be explained by salinity and environmental predictability, suggests that lines from highly predictable environments displayed higher salinity differences than those from lowly predictable environments. However, we did not detect a significant marginal effect of salinity ($R^2_{adj}$ = 0.79%; $P$ = 0.356) or $\rho^2$ × salinity interaction ($R^2_{adj}$ = 0.30%; $P$ = 0.562) on the overall DNA methylation pattern, suggesting the absence of overall DNA methylation changes in response to salinity or a lack of power to detect small variation being explained.

That we did not detect significant differentiation at the whole epigenome level does not preclude the existence of more localized epigenetic differences along the genome, so we also investigated regional changes in DNA methylation at a finer scale, by considering nonoverlapping 100 bp windows, hereafter denoted differentially methylated regions (DMRs). We detected 227 DMRs (with FDR < 0.05 under Benjamini–Hochberg (BH) adjustment of $P$ values and | diff-Methylation| > 20%) among the evolutionary treatments, as summarized by their environmental predictability $\rho^2$ (Fig 2B). We also detected $\rho^2$-specific DMRs between salinities within each evolutionary treatment, among the 14,357 total 100 bp regions (Fig 2B). Interestingly, populations that evolved under less predictable environmental fluctuations displayed the least number of DMRs between salinities ($n$ = 3), as compared to populations from intermediate ($n$ = 78) or high ($n$ = 29) environmental predictability (Fig 2B), indicative of reduced epigenetic plasticity. We then assessed whether changes in DNA methylation patterns in response to a given environmental challenge involved similar genomic regions in the different evolved lines. Comparison of the list of DMRs between salinities revealed no overlap across evolutionary conditions (Fig 2B), suggesting that evolution of plastic epigenetic responses involved modifications of methylations in distinct genomic regions in different treatments.

We next investigated variation in gene expression, by analyzing 32,718 transcripts through RNA-sequencing (RNA-seq). We obtained $5.62 \times 10^8$ 150 bp paired-end raw reads in total (S2 Table). As with DNA methylation, we detected a significant effect of the evolutionary treatments $\rho^2$ ($R^2_{adj}$ = 9.79%; $P$ < 0.001) on gene expression at the whole-transcriptome level. However, the assay salinity now explained the greatest part of variation in gene expression ($R^2_{adj}$ = 32.91%; $P$ < 0.001), indicating pervasive and highly significant plasticity. The $\rho^2 \times$ salinity interaction was also significant ($R^2_{adj}$ = 4.30%; $P$ = 0.029) (Fig 3A), indicating evolution of transcriptional plasticity. At a more local level, analyses of differentially expressed (DE) transcripts yielded similar results: The numbers of transcripts that were significantly DE (*Likelihood ratio test*, FDR < 0.05 and |Log$_2$FC| > 1) were highest for contrasts between salinities ($n$ = 4,283), followed by evolutionary treatments $\rho^2$ ($n$ = 1,315), and, finally, $\rho^2 \times$ salinity interaction ($n$ = 837). As for DMRs, populations that evolved in less predictable environments displayed fewer DE transcripts than populations that evolved in highly predictable environments (Fig 3B, $n$ = 2,638, 2,844, and 4,086 for $\rho^2$ = low, intermediate, and high, respectively). However, in contrast to what we found for DNA methylation, we observed substantial overlap among DE transcripts identified between salinities for the different evolved lines (Fig 3B), indicating that the plastic response to salinity largely involved transcriptional regulation of a common pool of genes, regardless of their evolutionary trajectory. Nonetheless, we still detected some $\rho^2$-specific DE transcripts between salinities (Fig 3B).

To confirm that differences in gene expression (and, to a lesser extent, DNA methylation) among salinities were due to plasticity, rather than resulting from putative strong selection taking place in the polymorphic populations during the short duration of the assay, we also sequenced isogenic populations—founded from a single, presumably haploid cell from the evolved populations—subjected to different salinities. Despite the genetic homogeneity of these population, we confirmed the salinity effect on both DNA methylation and gene expression (S1 Fig). Furthermore, we did not detect any significant differences in either DNA methylation ($P$ = 0.423) or gene expression ($P$ = 0.520) between isogenic lines and the experimental populations they originated from (S2 Fig).

We next performed Gene Ontology (GO) enrichment analysis, to assess gene functions involved in *D. salina* response to salinity changes and its evolution, using the functional genome annotation constructed by Leung and colleagues [23]. Transcript functional annotation analysis revealed that DE transcripts between salinities mostly involved genes associated with cellular components and molecular functions involving the chloroplast and protein

A.

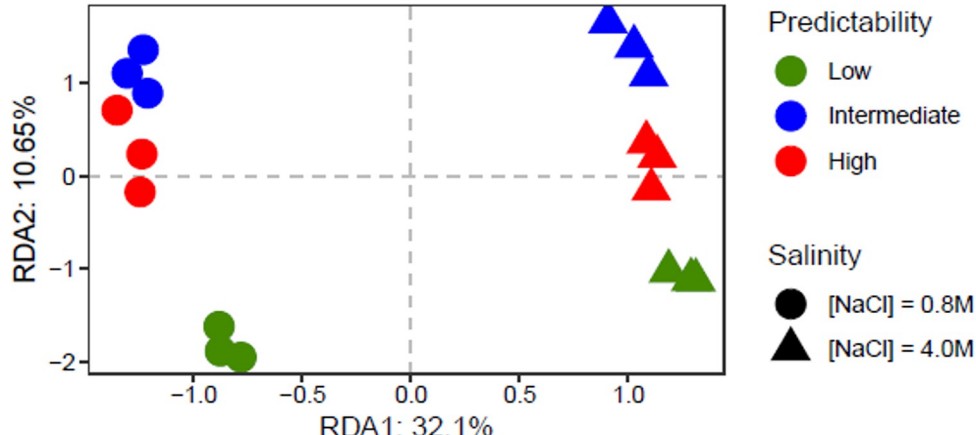

B.

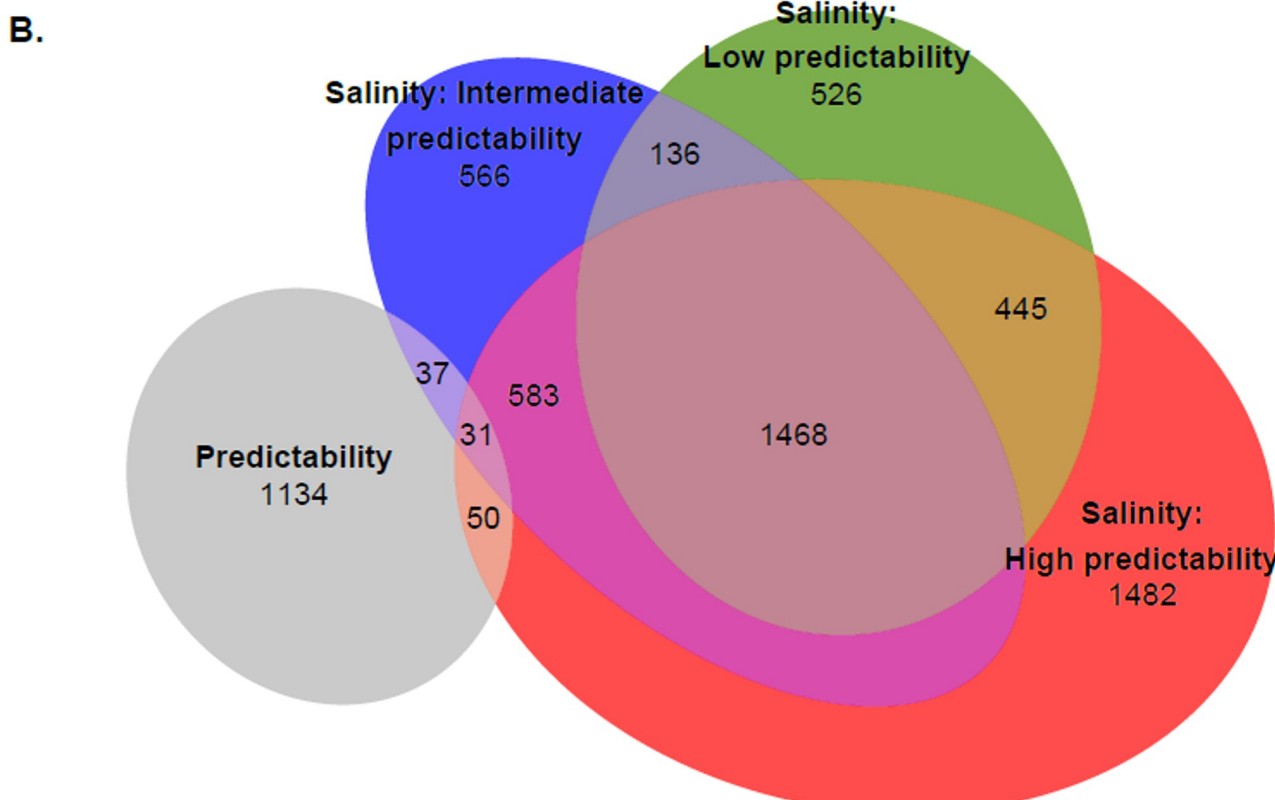

**Fig 3. Evolution and plasticity of gene expression. (A)** Variation of gene expression levels across evolved populations. RDA plot performed on gene expression levels according to evolution conditions (colors) and salinity during plasticity assay (shapes). **(B)** Distribution of DE transcripts. Venn diagram describing the numbers of DE transcripts across evolutionary conditions (Predictability, light gray) identified by performing LRT (FDR < 0.05) as implemented in *DESeq2*, and between salinities (Wald test, FDR < 0.05 after BH adjustment and $|log_2FC| > 1$) for 3 evolutionary conditions: low (green), intermediate (blue), and high (red) predictability of environmental changes. The raw data underlying this figure are available in the Figshare repository https://doi.org/10.6084/m9.figshare.21905670. DE, differentially expressed; LRT, likelihood-ratio test; RDA, redundancy analysis.

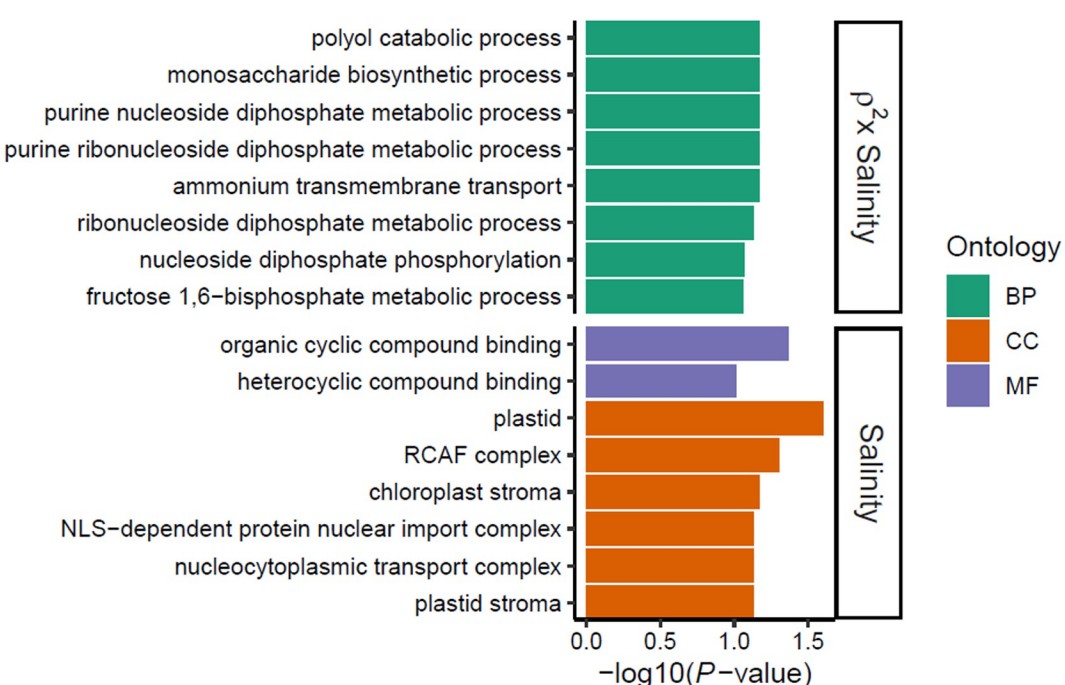

**Fig 4. Gene functions involved in salinity-induced plasticity and its evolution.** Enriched GO terms of identified DE transcripts between salinities (Salinity) and $\rho^2$-specific transcriptional response ($\rho^2 \times$ Salinity). GO categories included MF (purple), CC (orange), and BP (green) and were sorted by decreasing order of evidence within each category based on GO enrichment test *P* value (for FDR $\leq$ 0.1). No GO term enrichment with FDR $\leq$ 0.1 was detected for DE transcripts among evolutionary treatments ($\rho^2$). The raw data underlying this figure are available in the Figshare repository https://doi.org/10.6084/m9.figshare.21905670. BP, biological process; CC, cellular component; DE, differentially expressed; GO, Gene Ontology; MF, molecular function.

transport (Fig 4). While no GO term enrichment was detected for DE transcripts among evolutionary treatments (for FDR < 0.1), we found that $\rho^2 \times$ salinity interaction effects on gene expression essentially involved gene functions associated with different metabolic processes within the GO category "biological process" (Fig 4).

## 2.2 DNA methylation in *cis* was not associated with gene expression

We then asked whether DNA methylation could influence the expression of neighboring genes in *cis*. For each methylated cytosine, we searched for the nearest transcription start site (TSS). For this analysis, all cytosines associated to the same transcript were merged into a common gene-associated methylation region, following the same method used by Leung and colleagues [23]. Among the 3,064 methylated regions associated to a transcript, we detected 18 regions that were significantly differentially methylated among evolutionary treatments (including all comparisons of $\rho^2$ pairs), and 21 regions among salinities (for each evolutionary treatments $\rho^2$). However, only 4 of these 39 transcript-associated regions were associated with significant DE transcripts for the same comparison, and only one involved a comparison between salinities. This suggested that, should DNA methylation have an effect on gene expression plasticity, this effect must be acting on regulation in *trans*.

## 2.3 The degree of plasticity evolved consistently across levels

We detected significant effect of salinity ($R^2_{adj}$ = 2.51%; $P < 0.001$), evolutionary treatment $\rho^2$ ($R^2_{adj}$ = 4.13%; $P < 0.001$), and their interaction ($R^2_{adj}$ = 1.06%; $P < 0.001$) on cell morphology,

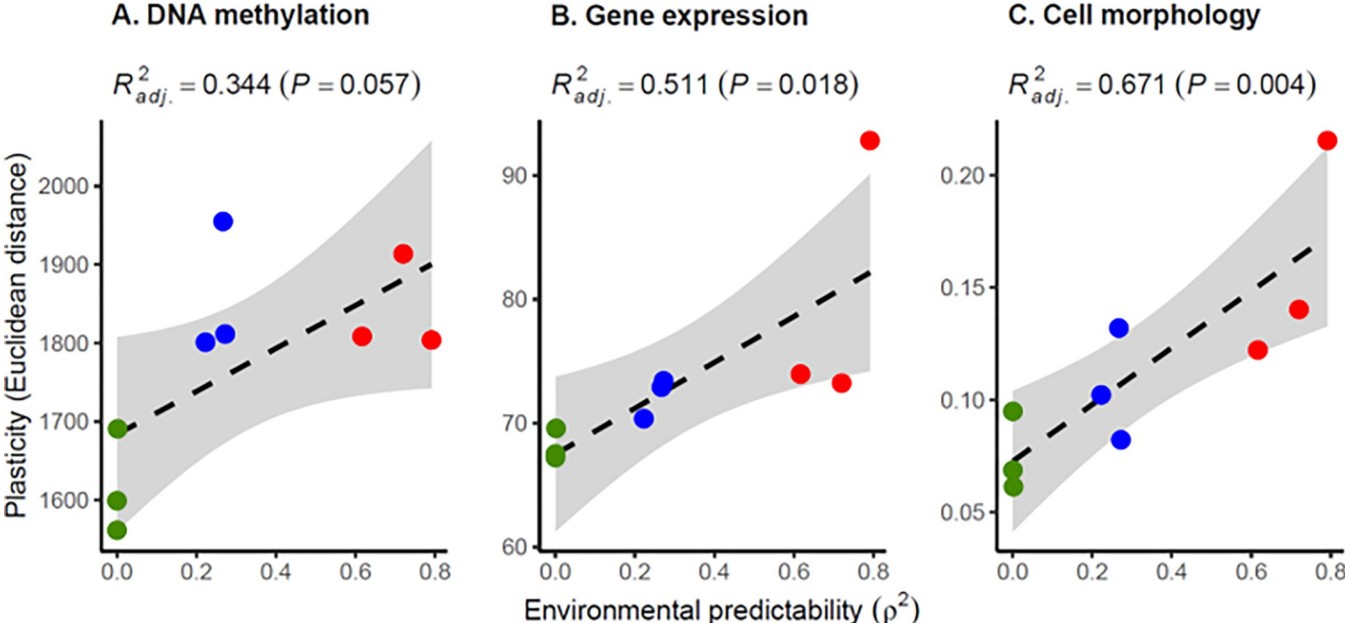

**Fig 5. Multilevel evolution of plasticity in response to environmental predictability.** The degree of plasticity of populations, as measured by the Euclidean distance between low and high salinities for (**A**) DNA methylation patterns at CpG context, (**B**) gene expression levels, and (**C**) cell morphology, is plotted against the predictability $\rho^2$ of the environmental fluctuations that these populations have experienced during experimental evolution. The dashed line is the regression slope, and the gray area represents the 95% confidence interval of the linear regression. The raw data underlying this figure are available in the Figshare repository https://doi.org/10.6084/m9.figshare.21905670.

thus providing independent replication of the results from Leung and colleagues [5]. To investigate the consistency in the direction of experimental evolution of plasticity across levels, we then quantified the overall degree of plasticity for DNA methylation, gene expression, and cell morphology, measured as the Euclidean distance between the multivariate means at low versus high salinities. When regressing the magnitude of plasticity against the environmental predictability of the experimental evolution treatment, we consistently found a positive relationship, with weak to moderate evidence (sensu [29]) for DNA methylation ($P$ = 0.057; Fig 5A), but strong evidence for gene expression ($P$ = 0.018; Fig 5B) and cell morphology ($P$ = 0.004; Fig 5C). Hence, populations that have experienced less predictable environments during experimental evolution have evolved reduced plasticity, not only for cell morphology (as shown by Leung and colleagues [5]), but also for DNA methylation and gene expression, two important molecular mechanisms potentially underlying phenotypic plasticity.

## 3. Discussion

We aimed to understand whether the evolution of plasticity in response to environmental predictability is a phenomenon that can be observed at different hierarchical levels. We therefore exposed experimental populations of the microalga *D. salina* to randomly fluctuating salinity with controlled predictability, before assessing how their plasticity evolved at 3 levels: DNA methylation, gene expression, and cell morphology. Our results highlighted important aspects of the molecular mechanisms of plasticity and its evolution.

### 3.1 Gene expression plasticity and its regulation

A wealth of studies have investigated the molecular underpinnings of phenotypic plasticity, and most of them have identified gene expression as a key mechanism of phenotypic changes

[10–14]. A crucial step in gene expression is transcription to mRNA, which explains a large fraction of variation in protein abundance [30] and, hence, in more integrated phenotype. Here, we found that salinity-induced transcriptional plasticity of *D. salina* involved largely overlapping DE transcripts among different populations, indicating that this species has specific genes to cope with salinity changes in its environments, consistent with its halotolerant ecology [31,32]. In addition, we detected significant $\rho^2 \times$ salinity interaction for gene expression, indicative of evolution of transcriptional plasticity with respect to environmental predictability. While only ca. 30% of the total transcripts were successfully assigned to at least one GO term [23], we were able to assess some gene functions involved in *D. salina* response to salinity. On the one hand, common salinity responses mostly involved genes associated with chloroplast and membrane-associated functions, like protein transport. On the other hand, $\rho^2$-specific salinity responses entailed biological processes necessary for the production of energy for various functions of the organisms [33–35]. From a functional standpoint, these results suggest that *D. salina* preserves its complex molecular machineries for coping consistently with osmotic stress, while evolutionary differentiation in salinity-induced plasticity may involve potential energy allocation trade-offs among different functions.

Regulation of gene expression involves different mechanisms, including epigenetic processes [36,37]. Here, we investigated one such epigenetic process, namely DNA methylation. We detected DMRs between salinities in every evolutionary treatment, confirming that DNA methylation is a process that could be involved in phenotypic plasticity in this species [23]. However, we failed to detect a correlation between DMRs and the expression of proximal genes, suggesting that methylation-based regulation of gene expression plasticity for this species may be driven by *trans*, instead of *cis*, regulatory changes [38,39]. Alternatively, mechanisms of gene expression regulation could be highly stress- and organism-dependent [40]. The time lapse (24 h) between osmotic changes and DNA methylation measurements in our study, despite being long enough to allow for epigenetic marks to be modified during DNA replication, could still be too short to enable detection of significant correlations between DNA methylation and gene expression in *D. salina* (e.g., [41]). The first responses to osmotic shock in *D. salina* may implicate homeostatic regulation, relying on mechanisms other than DNA methylation [42].

Exploring the genetic basis of phenotypic plasticity and its evolution requires the identification of genes whose expression varies across environments, as well as loci contributing to this variable degree of plasticity among genotypes. Interestingly, the $\rho^2$-specific DMRs for the 3 categories of evolutionary treatments did not display any overlap, suggesting that different regions in the genome were involved in the control of salinity-induced plasticity. This could be the result of epistatic interactions, where different loci could contribute to fine-tuning the phenotype toward its optimum [43–48].

## 3.2 Multilevel evolution of plasticity

To what extent do we expect evolution to take the same course across different biological levels, along the hierarchy of traits that relates the genotype to more integrated phenotypes, and fitness? This question has received increasing attention in the evolutionary literature, where it has mostly been expressed with respect to parallel/convergent evolution in response to a constant environmental challenge [49,50]. Theoretical and empirical work has established that the amount of parallelism at each biological level depends on the degree of redundancy in the mapping from one level to the next [46–48]. This, in turn, depends on the extent to which each item at a given level is connected to one or many items at the level above, e.g., how many traits a mutation can modify (pleiotropy) [51–53]. Because mappings along hierarchies of traits

are often redundant, we do not necessarily expect to find the same degree of parallelism at the level of genes as we find at the level of functional categories, organs, or highly integrated traits [49,50]. And reciprocally, the degree of parallelism across levels conveys information about the redundancies in mappings along hierarchies of traits.

Here, we applied a similar approach to a classic topic from theoretical evolutionary ecology: the evolution of phenotypic plasticity in response to environmental predictability [3–7]. Just like for evolutionary parallelism, we do not necessarily expect the evolution of plasticity in response to environmental predictability to be consistent across levels. Indeed, it was recently highlighted that the comparison of phenotypic plasticity (and its evolution) across levels can provide important information about constraints in the genotype-(environment)-phenotype map, including—but not restricted to—the abovementioned redundancies. For example, this can yield insights about the extent to which mutations and environments have similar influences on phenotypic variation, and which one is likely to drive evolution of the other [54,55]. Here, the significant DMRs and DE transcripts detected across evolutionary treatments can be interpreted as parallel evolution of DNA methylation and gene expression, respectively. Moreover, we not only found substantial plasticity at the levels of DNA methylation, gene expression, and cell morphology, but also that plasticity evolved in a consistent direction—and consistent with theoretical predictions—in response to environmental predictability across these levels. This response was, however, less pronounced for DNA methylation, which only displayed marginally significant relationship between total plasticity and predictability (Fig 5A), but with substantial variation in the number of salinity DMRs among evolutionary treatments (Fig 2B).

Such consistent evolution of plasticity across levels may indicate that many unmeasured integrated traits exhibit similar evolutionary responses to environmental predictability as those we have measured here (cell morphology), such that their transcriptomic (and, to some extent, epigenetic) basis is less subject to redundancies. Interestingly, in a study focusing on other lines from this same experiment [56], we found that plasticity of intracellular glycerol concentration—a major mechanism for osmoregulation in this species [57,58]—also evolved in response to environmental predictability, but in the opposite direction (higher plasticity in populations from unpredictable environments). However, this was mostly explained by the higher glycerol levels maintained by populations from unpredictable treatments (relative to those from predictable treatments) at intermediate and high salinities (but not at low salinity), which can be interpreted as a benefit of maintaining hyperoptimal phenotypes when fitness functions are highly asymmetrical [59], as shown in this species with respect to salinity transitions [24,56]. Alternatively, consistent evolution of plasticity across levels may indicate that selection operates, to some extent, independently at each of these hierarchical levels. For example, expression plasticity of a given gene may partly influences fitness by itself, rather than only via its effect on the plasticity of more integrated phenotypes. Investigating these questions would require mechanistic studies beyond the scope of this work. Nevertheless, our finding that plasticity can experimentally evolve at multiple levels in response to environmental predictability, and in a consistent direction, brings much needed empirical insights into the mechanisms underlying plasticity evolution, and highlights the usefulness of experimental evolution for investigating ambitious questions at the forefront of evolutionary biology.

## 4. Materials and methods

### 4.1 Experimental evolution conditions and plasticity assays

To investigate the evolution of plasticity in responses to environmental predictability at the molecular level, we analyzed different populations of the *D. salina* strain CCAP 19/15 that

evolved in a fluctuating environment for ca. 500 generations, from the experiment described in Rescan and colleagues [24] and Leung and colleagues [5]. Briefly, different populations starting from the same ancestor were exposed to randomly fluctuating salinity, with changes every 3 or 4 generations (i.e., twice a week, assuming one generation per day [31]). Each of these populations were subjected to independent random time series of salinity changes, over a continuous range. The time series were characterized by the same stationary mean ($\mu = 2.4$ M [NaCl]) and variance ($\sigma = 1$) but differed in how salinity at a given time depends on the salinity prior the transfer (i.e., temporal autocorrelation $\rho$ of salinity). The predictability of salinity changes for a given time series was assessed by $\rho^2$, the proportion of temporal variance in salinity explained by the previous salinity, prior to the latest transfer (see Rescan and colleagues [24] and Leung and colleagues [5] for detailed protocol). We specifically analyzed 9 populations from 3 different target autocorrelation treatments ($\rho = 0$, $-0.5$, and $0.9$; 3 populations per autocorrelation treatments). These autocorrelation treatments correspond to low ($\rho^2 = 0$), intermediate ($\rho^2 = 0.25$), and high ($\rho^2 = 0.81$) predictability of environmental changes. While similar levels of morphological plasticity were previously found between populations that evolved in intermediate predictability changes but with different autocorrelation treatments ($\rho = -0.5$ and $\rho = 0.5$; Leung and colleagues [5]), here we chose the negative autocorrelation to include a larger range of $\rho$.

Plasticity assays were performed following the protocol described in Leung and colleagues [5]. We first acclimatized all evolved lines during 10 days in the same environmental conditions ([NaCl] = 2.4 M) to ensure that all cells were in similar physiological states and at similar population densities at the beginning of the phenotypic assays. Cells grew in suspension flasks containing artificial seawater with additional NaCl to reach the required salinity, complemented with 2% Guillard's F/2 marine water enrichment solution (Sigma; G0154–500 ML), and incubated at a constant temperature 24°C with a 12:12 h light/dark cycle with a 200-μmol $m^{-2}$ $s^{-1}$ light intensity. Target salinity was achieved by mixing the required volumes of hypo- ([NaCl] = 0 M) and hyper- ([NaCl] = 4.8 M) saline media, accounting for the salinity of the inoculate. At the end of the acclimation step, we transferred ca. $1 \times 10^5$ cells $mL^{-1}$ of each populations to low ([NaCl] = 0.8 M) and high ([NaCl] = 4.0 M) salinities, for a total volume of 250 mL. After 24 h, following the salinity changes, we harvested the cells by centrifugation at 5,000 rpm for 15 min at room temperature, and cell pellets were stored at −80°C until acid nucleic extraction.

We also measured intrinsic structural parameters of cells by passing a subsample of 150 μL of each populations through a Guava EasyCyte HT flow cytometer (Luminex Corporation, TX, USA), also following the protocol described in Leung and colleagues [5]. We specifically assessed the environment-specific cell morphology using the Forward Scatter (FSC) and Side Scatter (SSC) as proxies for cells size and complexity (cytoplasmic contents) [60], respectively, and fluorescence emission at 695/50 nm band pass filter (Red-B) values for chlorophyll content [61]. The cell morphology matrix consisted of values for these 3 parameters (FSC, SSC, and Red-B), for 150 randomly sampled cells identified as alive *D. salina* for each populations.

## 4.2 Sample preparation, sequencing, and bioinformatic preprocessing

To investigate the molecular mechanisms involved in osmotic stress responses, we performed whole-transcriptome shotgun sequencing (RNA-seq) for the comparison of gene expression levels, and whole-genome bisulphite sequencing (WGB-seq) for the comparison of DNA methylation variation among the 9 evolved lines and 2 environmental conditions (hypo- and hyperosmotic). The different lines started from potentially genetically diverse population and could thus be polymorphic at the end of experimental evolution. To assess whether genetic

variation within population affected the observed plasticity levels for gene expression or intra-population DNA methylation variation, we also founded 3 isogenic populations (one per auto-correlation treatment) started from a single cell using cells-sorting flow cytometry (BD FACSAria IIu; Biosciences-US). As *D. salina* is haploid, we expected all derived cells of a given population to be genetically identical. Each of these isogenic populations were also subjected to the plasticity assay described above.

Total RNA extraction and purification of 24 samples ((9 lines + 3 isogenic populations) × 2 salinities) was carried out using Nucleozol, following Macherey Nagel's protocol, and whole genomic DNA was isolated according to the phenol-chloroform purification and ethanol pre-cipitation method of Sambrook and colleagues [62]. Library construction (TruSeq RNA Library Preparation kit for RNA-seq and Swift Bioscience Accel-NGS Methyl-Seq DNA library Kit for WGB-seq) and high-throughput sequencing steps (Paired-End (PE) 2 × 150 bp, Illu-mina HiSeq) were performed by Genewiz (Leipzig, Germany). We performed all the bioinfor-matic preprocessing analyses using publicly available software implemented in the European UseGalaxy server [63].

**Methylation calling.** The WGB-seq raw reads were checked for quality using *FastQC*. Adapter and low-quality sequences were then trimmed using *Trim Galore*! Version 0.4.3.1. As specified by the Accel-NGS Methyl-seq Kit manual, additional 15 bp and 5 bp were also trimmed at the 5′ and 3′ extremity, respectively, to remove the tail added during library preparation and thus avoiding nonquality-related bias. Mapping was performed on the same references genomes as in RNA-seq analyses, using *Bismark Mapper* version 0.22.1 [64]. Only uniquely mapping reads were retained and PCR duplicates were removed using *Bismark Deduplicate* tool. We then extracted the methylation status from the resulting alignment files using *MethylDackel* (Galaxy Version 0.3.0.1), where only cytosines covered by a minimum of 10 reads in each library were considered, and with the option of excluding likely variant sites (i.e., minimum depth for variant avoidance of 10×, and maximum toler-ated variant fraction of 0.95). In a previous study, we showed that cytosines at CpG context displayed the highest methylation levels (and variation thereof) in *D. salina* ([23]; S2 Table). Furthermore, CpG-methylations have been suggested to play a role in gene regulations [65–67] and proposed as a molecular mechanisms underlying phenotypic plasticity [15,17]. We thus investigated the genomic DNA methylation patterns in response to salinity across the different evolutionary treatments for cytosines at the CpG context. The high bisulfite con-version rate (>99%) was assessed by Genewiz, by spiking in unmethylated lambda DNA in 3 randomly chosen libraries.

**Gene expression analyses.** The RNA-seq raw reads were checked for quality using *FastQC* version 0.72 [68] and subjected to adapter trimming and quality filtering using *Trim Galore*! version 0.4.3.1 [69]. Additional 12 bp and 3 bp were also removed at the 5′ and 3′ extremity, respectively, to avoid bias not directly related to adapter sequences or basecall quality according to *FastQC* outputs, and only reads with a minimum length of 50 bp were retained. We used the reference nuclear (Dunsal1 v. 2, GenBank accession: GCA_002284615.2), chloroplastic (GenBank accession: GQ250046), and mitochondrial (GenBank accession: GQ250045) genomes of *D. salina* strain CCAP 19/18 (closely related to CCAP 19/15 used here) for trimmed reads alignment, using *HISAT2* version 2.1.0 [70] with default parameters for PE reads and spliced alignment option. We finally quantified the number of reads per transcript with *FeatureCounts* version 2.0.1 [71] using the align-ment files from *HISAT2* and the de novo transcript annotation produced for this species from Leung and colleagues [23] using StringTie v. 2.1.1 [72] and including the evolved lines of the current study.

### 4.3 Statistical analyses

**Differential DNA methylation and gene expression.** We used Bioconductor's *methylKit* package [73] to identify DMRs, i.e., nonoverlapping 100 bp windows with methylation levels that varied significantly among evolutionary treatments or between salinities within each treatments. The significance of calculated methylation differences was determined using Fisher's exact tests. We used the BH adjustment of *P* values (FDR < 0.05) and methylation difference cutoffs of 20%. Similarly, the differential gene expression analyses were performed with the Bioconductor's package *DESeq2* version 1.30.1 [74]. We identified DE transcripts among the evolutionary treatments, salinity, and their interaction, by building a general linear model as implemented in *DESeq2*. We used the Wald test when comparing 2 salinities, and transcripts with FDR < 0.05 (*P* values after BH adjustment) and $|\log_2 FC| > 1$ were considered as DE. Significance of evolution (predictability $\rho^2$) and evolution of plasticity ($\rho^2 \times$ salinity interaction) were assessed using likelihood ratio tests (LRTs) comparing models with and without the corresponding terms [74].

We then used the GO assignments from Leung and colleagues [23] to classify the functions of *D. salina* transcripts and to functionally annotate the identified DE transcripts. Enriched GO terms of the DE transcripts were identified using the 'classic' algorithm from the *topGO* R package [75] and based on *p*-value generated using Fisher's exact method, for the 3 GO categories, i.e., molecular function, cellular component, and biological process. GO terms were then sorted by decreasing order of evidence within each category, based on the GO enrichment test *P* value after BH adjustment, and we showed the most probable gene function candidates with a threshold of FDR $\leq$ 0.1.

**Evolution of the degree plasticity at multiple levels.** Since phenotypic plasticity can evolve as an adaptation to fluctuating and predictable environment [3–7], we wished to quantify to what extent environmental predictability during experimental evolution contributed to key mechanisms underlying phenotypic plasticity. We first applied a RDA [76], computed with the function *rda()* from the *vegan* R package [77], to quantify the proportion of the total epigenetic and gene expression variation that are significantly explained by the evolutionary treatments (predictability $\rho^2$) and assay environment (salinity). We performed a RDA using the table of DNA methylation levels of 100 bp windows regions, or the table of *rlog* transformed transcript count, as response variables, and the evolutionary treatments (predictability $\rho^2$), salinity, and the $\rho^2 \times$ salinity interaction, as explanatory variables, with population identity as covariates to account for paired samples between salinities. We treated the target $\rho^2$ (low, intermediate, or high) and salinity (low or high) as categorical variables. We quantified the contribution to the total genetic variation using the adjusted $R^2$ and tested its significance by ANOVA-like permutation tests using 999 randomizations of the data [78]. For both RNA-seq and WGB-seq data, we presented the results of the RDA ordinations as a biplot, to visualize the variation among samples along its major axes. We performed all multivariate statistical analyses with the *vegan* R package [77].

To investigate how the overall degree of plasticity at multiple levels evolved in our experiment, we followed the protocol described in Leung and colleagues [5], which we here applied to DNA methylation level, gene expression, and cells morphological variation. (Note that the morphological measurements were replicated here, rather than just reproducing values from Leung and colleagues [5].) First, we assessed the degree of plasticity of each experimental population by computing the Euclidean distance between the multivariate means from the CpG-methylation level table, the *rlog* transformed transcript count table, or the raw cells morphological data measured, at low ([NaCl] = 0.8 M) versus high ([NaCl] = 4.0 M) salinity. The realized autocorrelation of a given time series can vary, to some extent, from its long-term

stationary expectation, because of the randomness of the stochastic process in finite time. For a more quantitative relationship between plasticity and environmental predictability, we thus used the realized (rather than the target) $\rho^2$ as index of environmental predictability (as also done in [5]). For each time series, we thus calculated the realized environmental autocorrelation $\rho$ as the correlation between salinities at 2 subsequent transfers. We then tested whether the degree of plasticity evolved according to environmental predictability, by regressing the Euclidean distance of plastic changes against the realized $\rho^2$. The linear regression $t$ test was applied to determine whether the slope of the regression line differs significantly from zero.

## Supporting information

**S1 Table. WGB-seq mapping statistics.** For each sample, population ID, realized $\rho^2$, and salinity are detailed in S2 Table. We used a genome size of 350 M bp to estimate depth of coverage. Total C's analyzed were after reads deduplication and number of methylated C's and percentage of methylation are given by cytosine context.
(XLSX)

**S2 Table. RNA-seq mapping statistics.**
(XLSX)

**S1 Fig. Salinity effect on DNA methylation and gene expression in isogenic populations.** We founded 3 populations from single isolated cells from 3 evolved populations, following the protocol in Leung and colleagues [5]. As *D. salina* is haploid, a population founded from a single cell is expected to be isogenic. (**A**) Heat-maps of WGB-seq analysis for DMRs between salinities ($n = 27$). Each row represents a DMR, and column names are the population identity. Relative DNA methylation levels vary from blue (under-methylated) to red (over-methylated), as shown on the right-hand side of the heat-maps. Dendrograms on the top result from a hierarchical clustering analysis using the Euclidean distance of DNA methylation level among populations. (**B**) Volcano plot illustrating significant (for FDR < 0.05 and $|Log_2FC| > 1$) and nonsignificant DE transcripts between salinities as red and gray points, respectively. Salinity effect was assessed by comparing three isogenic populations (i.e., found from a single cell). The raw data underlying this figure are available in the Figshare repository https://doi.org/10.6084/m9.figshare.21905670. DE, differentially expressed; DMR, differentially methylated region; WGB-seq, whole-genome bisulfite sequencing.
(PDF)

**S2 Fig. Comparison of isogenic and experimental populations.** Principal component analysis (PCA) of (**A**) DNA methylation and (**B**) gene expression levels among isogenic (open symbols) and experimental non-isogenic (filled symbols) populations. The raw data underlying this figure are available in the Figshare repository https://doi.org/10.6084/m9.figshare.21905670.
(PDF)

## Acknowledgments

We thank Rachel Steward for useful feedback on previous versions of this manuscript.

## Author Contributions

**Conceptualization:** Christelle Leung, Daphné Grulois, Luis-Miguel Chevin.

**Formal analysis:** Christelle Leung.

**Funding acquisition:** Christelle Leung, Luis-Miguel Chevin.

**Methodology:** Christelle Leung, Daphné Grulois, Leandro Quadrana.

**Supervision:** Luis-Miguel Chevin.

**Writing – original draft:** Christelle Leung, Luis-Miguel Chevin.

**Writing – review & editing:** Christelle Leung, Daphné Grulois, Leandro Quadrana, Luis-Miguel Chevin.

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
