## [Editor Report · Decision Letter 0]

28 Oct 2022

Dear Dr Leung, 

Thank you for submitting your manuscript entitled "The molecular basis of phenotypic plasticity evolves in response to environmental predictability" for consideration as a Research Article by PLOS Biology.

Your manuscript has now been evaluated by the PLOS Biology editorial staff, as well as by an academic editor with relevant expertise, and I'm writing to let you know that we would like to send your submission out for external peer review.

Once your full submission is complete, your paper will undergo a series of checks in preparation for peer review. After your manuscript has passed the checks it will be sent out for review. To provide the metadata for your submission, please Login to Editorial Manager (https://www.editorialmanager.com/pbiology) within two working days, i.e. by Nov 01 2022 11:59PM.

Kind regards,

Roli Roberts

Roland Roberts, PhD

Senior Editor

PLOS Biology

rroberts@plos.org

---

## [Decision Letter · Decision Letter 1]

5 Jan 2023

Dear Dr Leung,

Thank you for your patience while your manuscript "The molecular basis of phenotypic plasticity evolves in response to environmental predictability" went through peer-review at PLOS Biology. Your manuscript has now been evaluated by the PLOS Biology editors, an Academic Editor with relevant expertise, and by two independent reviewers. We had recruited a third reviewer, but they failed to deliver their comments on time.

In light of the reviews, which you will find at the end of this email, we are pleased to offer you the opportunity to address the comments from the reviewers in a revision that we anticipate should not take you very long. You'll see that reviewer #1 is broadly positive, and his main complaint is that you unnecessarily oversell it by downplaying prior work; he also thinks he sees an interaction that you don’t mention, and asks for one minor analysis (to compare replicates). Reviewer #2 just has a few minor queries. We will then assess your revised manuscript and your response to the reviewers' comments with our Academic Editor aiming to avoid further rounds of peer-review, although might need to consult with the reviewers, depending on the nature of the revisions.

IMPORTANT: Please address the following points:

a) Please attend to the requests from the reviewers. The Academic Editor asked me to emphasise the following point, raised by reviewer #1: "... one additional bit of analysis has to do with repeatability. There were three replicates for each treatment. Were the same molecular responses seen in each?"

b) Please provide a blurb, according to the instructions in the submission form.

c) Please comply with our Data Policy; specifically, we need you to supply the numerical values underlying Figs 2AB, 3AB, 4, 5ABC, S1AB, either as a supplementary data file or as a permanent DOI’d deposition (e.g. Figshare, Dryad, Zenodo, etc.).

d) Please cite the location of the data clearly in all relevant main and supplementary Figure legends, e.g. “The data underlying this Figure can be found in S1 Data” or “The data underlying this Figure can be found in https://doi.org/XXXX”

**IMPORTANT - SUBMITTING YOUR REVISION**

*Resubmission Checklist*

*Published Peer Review*

*PLOS Data Policy*

*Blot and Gel Data Policy*

Sincerely,

Roli Roberts

Roland Roberts, PhD

Senior Editor

PLOS Biology

rroberts@plos.org

REVIEWERS' COMMENTS:

Reviewer #1:

[identifies himself as Samuel Scheiner]

This manuscript reports on a deep dive into the molecular bases of an evolved response to an artificial selection experiment by phenotypic plasticity. In general the experiment is well designed and carried out. I have no complaints with what was done, although I do have one suggested additional bit of analysis. 

However, I do have one complaint with the way that the work is being framed. According to the paper: "However, we still lack critical empirical evidence on the extent to which selection on phenotypic plasticity cascades down from higher phenotypic levels to their underlying molecular basis." We have decades of information that plasticity has a heritable genetic basis and that selection on plasticity can result in an evolutionary response. We even have a growing list of studies of the molecular bases of plasticity. So to say that we "lack critical empirical evidence" is incorrect. 

What we do lack, and what this paper clearly explores, is the ways that different types of molecular changes respond to different intensities of selection for plasticity. The use of three different levels of predictability, combined with the detailed molecular and cellular measurements makes this a very valuable study. There is no need to oversell it. 

I do have one question about the results. For the analysis of DNA methylation, the text states that there was no rho-salinity interaction (line 110). But looking at Fig 2A, there looks like there is a pretty strong interaction (no salinity difference at low predictability and a large difference at predictability). Perhaps I am misunderstanding the figure, or there is a problem with low power as the total variation being explained is very small. Whatever the explanation, something needs to be said in the text as other readers are likely to have the same reaction as me. Or perhaps you can add confidence areas around the points.

The one additional bit of analysis has to do with repeatability. There were three replicates for each treatment. Were the same molecular responses seen in each? In figures 2 and 3 the replicates appear to cluster closely, implying repeatability, but something should be said explicitly about this issue. If there were differences, were they coordinated across levels? (I will admit that I am not sure exactly how that could be analyzed or shown.) If you can add this, it would help deepen the entire story.

Minor items

Lines 81-98: This material should be in Methods.

Lines 108-111: Besides just stating which terms were significant, it would helpful to add some sort of explanation about what each term means with respect to the main questions being posed. Or perhaps, this could be framed in the Introduction.

Line 128: change: "total (Table S2). As [with]"

Line 154: change "involved genes associated [with] cellular components"

Line 157: change "[found] that"

Line 158: insert "functions associated [with] different metabolic processes"

Line 160 change "DNA methylation in cis was not associated"

Line 168: change "regions were associated [with] significant DE transcripts"

Line 177: "plasticity for each of them, measured" It is unclear what "them" refers to.

Line 179-181: As above, it would be useful to provide some sort of explanation of the meaning of this ordering.

Lines 188-189: delete "We [] therefore exposed experimental"

Line 197: "higher phenotype" I am not generally in favor of the notion of higher and lower in this context. Perhaps, "more integrated".

Line 199: change "that this species [has] specific genes"

Line 200: change "Salinity [for] gene"

Line 204: change "involved genes associated [with]"

Line 208: change "molecular machineries [for coping] consistently with"

Line 216: insert "suggesting that [the] regulation of gene"

Line 219: insert "DNA methylation measurement[s]"

Line 220: insert "significant correlation[s] between DNA"

Line 222: change "relying on [mechanisms other] than DNA"

Line 228: change "could contribute to [fine-tuning] the"

Line 235: insert "work has [e]stablished that the amount"

Line 238: change punctuation "level above[,] for instance,"

Line 247: change "across levels can [provide important]"

Line 248: insert "the genotype-(environment)-phenotype [map]"

Line 249: change "this can yield insights [about] the extent"

Line 262: insert "For instance, expression [of] plasticity of a given gene"

Line 286: insert and change "similar level[s] of morphological plasticity [were] found between"

Line 288: I do not understand what is meant here. This is the first mention of any sort of negative autocorrelation.

Line 290: insert "the protocol descri[b]ed in Leung"

Line 318: change "population affect[ed] the observed plasticity"

Line 323: delete "plasticity assay [] described above"

Line 406: change "data was [replicated here], rather than"

Reviewer #2:

[identifies himself as Carl Schlichting]

This study examines patterns of methylation and transcription for 9 lines of Dunaliella subjected to Randomly fluctuating salinity for 500 generations; three regimes produced differences in autocorrelation (i.e., predictability of 0.00, 0.25 and 0.81. Lines were subsequently exposed to 0.8M or 4.0M salinity to assess plasticity and gene expression.

Results suggest that lack of predictability has the predicted effect of reducing plasticity.

Results of analysis of methylation and transcription patterns are similar, but not congruent: DMRs are only affected by evolutionary history; DEs affected mostly by salinity level, with a significant EH*salinity interaction.

A very solid study, with robust methods and analyses.

Comments

The authors state (L 141) that there is "great overlap among DE transcripts identified between salinities for the different evolved lines indicating that the plastic response to salinity largely involved transcriptional regulation of a common pool of genes". By my calculations this is only about 15% - there were 70% more DEs specific to EHs than those found in all EHs. Even if you add in all overlaps the numbers are almost the same.

L 155 - No GO term enrichment with FDR ≤ 0.1 was detected for DE transcripts among evolutionary treatments - any ideas about why this is?

L 215 - "we failed to detect a correlation between DMRs and the expression of proximal genes"

 I couldn't find this documented in the results anywhere - please include

L 323 - Did you compare isogenic lines with experimental lines?

Para at L 386 - I couldn't follow the description of the RDA analysis. RDA/PCA typically doesn't partition effects into explanatory variables. Did you use the RDA output as an input into ANOVA?

L 288, garbled

A couple of instances of Euclidian rather than Euclidean

New ref:

March-Salas, M., J. F. Scheepens, M. van Kleunen, and P. S. Fitze. 2022. Precipitation predictability affects intra- and trans-generational plasticity and causes differential selection on root traits of Papaver rhoeas. Frontiers in Plant Science 13:998169.

---

## [Editor Report · Decision Letter 2]

10 Feb 2023

Dear Dr Leung,

Thank you for your patience while we considered your revised manuscript "The molecular basis of phenotypic plasticity evolves in response to environmental predictability" for publication as a Research Article at PLOS Biology. This revised version of your manuscript has been evaluated by the PLOS Biology editors and the Academic Editor.

Based on our Academic Editor's assessment of your revision, we are likely to accept this manuscript for publication, provided you satisfactorily address the following data and other policy-related requests:

IMPORTANT - Please attend to the following;

a) Please change the Title to something more explicit and informative. We suggest: "Experimental evolution of a halophilic alga reveals that phenotypic plasticity evolves at multiple biological levels in response to environmental predictability" - this includes both the experimental system and some idea of the findings.

b) Please include the species name clearly in the Abstract.

c) Many thanks for providing the data in the Figshare deposition, and for citing its URL correctly in the Figure legends. However, we could not access your deposition in order to check your compliance with our data policy - please could you give us access, either by setting the deposition live, or by sending us a confidential reviewer link?

d) I note that you mention the reviewers in the Acknowledgements. While we appreciate the sentiment, this is against PLOS policy, so please could you remove this?

e) Please note that per journal policy, we do not allow the mention of "data not shown", "personal communication", "manuscript in preparation" or other references to data that is not publicly available or contained within this manuscript. If you have such mentions, please either remove mention of these data or provide figures presenting the results and the data underlying the figure(s).

We expect to receive your revised manuscript within two weeks. 

*Published Peer Review History*

*Press*

Sincerely,

Roli Roberts

Roland Roberts, PhD

Senior Editor,

rroberts@plos.org,

PLOS Biology

---

## [Editor Report · Decision Letter 3]

20 Feb 2023

Dear Dr Leung,

Thank you for the submission of your revised Research Article "Phenotypic plasticity evolves at multiple biological levels in response to environmental predictability in a long-term experiment with a halotolerant microalga" for publication in PLOS Biology. On behalf of my colleagues and the Academic Editor, Csaba Pál, I'm pleased to say that we can in principle accept your manuscript for publication, provided you address any remaining formatting and reporting issues. These will be detailed in an email you should receive within 2-3 business days from our colleagues in the journal operations team; no action is required from you until then. Please note that we will not be able to formally accept your manuscript and schedule it for publication until you have completed any requested changes.

Sincerely, 

Roli Roberts

Senior Editor

PLOS Biology

rroberts@plos.org